# Institutional Feasibility of Managed Aquifer Recharge in Northeast Ghana

**Lydia Kwoyiga \*** and **Catalin Stefan**

Department of Hydrosciences, TU Dresden, 01069 Dresden, Germany; catalin.stefan@tu-dresden.de
*   Correspondence: lydia.kwoyiga@tu-dresden.de

**Abstract:** As part of global efforts to address the challenges that are confronting groundwater for various purposes (including irrigation), engineering methods such as Managed Aquifer Recharge (MAR) have been adopted. This wave of MAR has engulfed some parts of Northern Ghana, characterized by insufficient groundwater for dry-season irrigation. Inspired by the strides of these schemes, the paper assesses the institutional feasibility of MAR methods in the Atankwidi catchment where dry-season farmers may lose their source of livelihood due to limited access to groundwater. We used both primary and secondary data, together with policy documents, to address the following questions: (i) What provisions and impacts formal government institutions had for MAR, and; (ii) what catchment-level institutions exist which may influence MAR. The results show that formal government institutions do not prohibit the adoption of MAR in the country. Among these institutions, it is realized that laws/legislative instruments provide sufficient information and support for MAR than policies and administrative agencies. Moreover, catchment-level institutions which are informal in the form of taboos, rules, norms, traditions, and practices, together with local knowledge play a significant role as far as groundwater issues in the catchment are concerned, and are important for the adoption of MAR methods.

**Keywords:** Managed Aquifer; Recharge; Groundwater; Institutions; Ghana

## 1. Introduction

Sustainability of water resources for any livelihood activity can be achieved through groundwater engineering methods, such as Managed Aquifer Recharge (MAR). This is because over the years, MAR has proved its ability to replenish aquifers and augment groundwater supplies. This is seen in its wider application where a global inventory puts MAR applications at 1200 schemes; a solution to addressing water scarcity under different climatic, geographic, and socioeconomic conditions [1]. Citing some examples, Page et al. [2] demonstrated that MAR is contributing to achieving sustainable water management, especially in urban areas, as it is also able to store water from different sources. It is, therefore, not surprising that MAR is considered to be a tool capable of promoting groundwater adaptation to climate change and its impacts [3].

Besieged by limited groundwater in some parts of northern Ghana, particularly in the dry season for farming, artificial methods of boosting groundwater resources which include MAR have been adopted [4]. Among the various techniques of MAR implemented are Aquifer Storage and Recovery (ASR) and Pit Infiltrations; both referring primarily to techniques of getting water infiltrated, as classified by International Groundwater Resources Assessment Centre IGRAC [5]. These schemes have offered rural farmers in this part of the country the opportunity to undertake dry-season irrigation in the face of challenging water conditions. Discussions relating to similar schemes are well-documented in places like India [6].

In the Atankwidi catchment in northern Ghana, the population depends almost entirely on groundwater resources. Most smallholder irrigators dig wells in the river bed and adjacent low-lying areas near rivers to exploit groundwater to meet their irrigation water needs. As part of the farming conditions, farmers are expected to refill these wells after every dry-season farming session, which makes the activity difficult. Also, Ghana designed a Riparian Buffer Zone Policy in 2011 [7], which was meant to create vegetative buffers for the preservation and functioning of the country's water bodies and vital ecosystems. When this policy is implemented, these famers will be without any livelihood activity. At the peak of the dry season, these farmers experience water scarcity, for which Barry et al. [8] noted that they will need better technology in order to access groundwater at deeper depths. These farmers' contributions to irrigation in the Upper East region is significant in that Dittoh et al. [9] noted them to be more than those engaged in surface water irrigation. Unfortunately, in recent times, the available groundwater for this activity have been insufficient.

Associated factors, such as increasing population and what Laube et al. [10] discussed as favourable conditions for the dry season (e.g., better infrastructure, the presence of a tomato factory etc.) are further increasing the demand for groundwater in general in the catchment. There are also reports of the incidence of fallen groundwater tables already in the northern part of Ghana, including Atankwidi [11,12] and these also have implications for groundwater availability for dry-season irrigation.

Apart from the aforementioned water challenges identified in the Atankwidi catchment, it has been documented that Ghana will become water-stressed by 2025. Climate change will further exacerbate the situation, as it will bring a reduction in groundwater recharge of 5–22% for 2020 and 30–40% for 2050 [13]. In a study on the Volta Basin about the potential impacts of climate change on subsurface and base flow for groundwater resources, the results show a reduction in recharges in the year 2020 of 17%, 5%, and 22% for Pra, Ayensu, and the White Volta, respectively, while for 2050 these values will increase to 29%, 36%, and 40% for the representative basins [14].

In view of the prevailing limited water conditions and those anticipated in the catchment, some farmers have abandoned farming or adopted measures which are either expensive or have little returns. For instance, Kwoyiga and Stefan [15] brought to the fore that in order to continue with dry-season farming, some irrigators have adopted strategies such as adopting crops that depend less on groundwater as a way of coping with limited groundwater. Unfortunately, some of these strategies do not yield better returns to the farmers.

Considering the importance of dry-season groundwater irrigation, especially in the Atankwidi catchment [16] and guided by the prospects of the emerging MAR schemes in the same northern Ghana, it will be prudent to assess the feasibility of MAR to boost groundwater for irrigation in the dry season. To make this feasible, it is realised that institutions play a crucial role in achieving MAR implementation. It is, therefore, not surprising that Gale [17] stated that even when the hydrological and hydrogeological parameters of a recharge scheme are favourable, consideration also needs to be given to institutions. Asano and Cotruvo [18] lamented that "the lack of guidelines governing artificial recharge of groundwater is currently hampering the implementation . . . of groundwater recharge operations".

Nonetheless, it is observed that while MAR activities are taking shape in Ghana, research relating to it at the moment focus on Geographic Information System GIS tools for mapping and technologies [4]. To the best of our knowledge, the institutional aspect of MAR is yet to receive attention. Even at the global level, it is realized that limited studies focus on the institutional aspect of MAR. For instance, Dillon et al. [3] provided a catalogue of eleven papers, and unfortunately, of their contributions to the increase of MAR application, only one paper discussed the role of institutions. Specifically in regard to Africa, though artificial recharge methods date back to the 1900s, existing literature largely focus on aquifer characterization and maps production [19]. This paper, therefore, assesses the scope of the existing formal government institutions and their implications on the adoption of MAR for irrigational purposes in the Atankwidi catchment.

It is realized that institutions define the kind of requirements or considerations (e.g., entitlement to share a source of water, a permit to construct a well, approval to recharge water to an aquifer, etc.) that

need to be met for MAR projects to be approved. These institutions also define regulatory frameworks to cover issues like water rights. They further define agencies or organizations that may be responsible for the management and operations of the project [20]. Apart from legal and regulatory issues, land rights, demand management of groundwater, and the conjunctive use of surface and groundwater are also given attention by institutions [17]. Mechlem [21] exemplified the situation by stating that prior authorization is a precondition for the success of MAR projects. Groundwater instruments like licenses are also needed to abstract, store, and recharge water with legislation dovetailing the conditions for the operation of such a project. Drawing examples from Arizona, Megdal and Dillon [22] note *inter alia* that the success of MAR can be attributed to these arrangements where proposed projects are subjected to rigorous scrutiny based on permit requirements, which include specifications of the kind of MAR projects that should be introduced. The focus here is, therefore, on the technical aspect, approval conditions, regulatory structures, and agencies/organizations that may affect MAR activities in the Atankwidi catchment.

The second objective of this paper looks at the informal institutions at the catchment level and how they may influence the implementation and management of MAR projects for dry-season farming. Page et al. [2] noted that apart from the hydrogeology, topography, and, hydrology, sociocultural and regulatory factors are crucial when choosing a suitable MAR site. Gale [17] opined this by admitting that the success of MAR was also dependent on the role of the local or rural people. Information from such people and their local environment could facilitate planning in terms of the area/community selection, participation responsibilities (construction), and technical options. Dillon [23] shared a similar view that the social environment is a factor that may largely determine the extent to which MAR can achieve its potential for water supplies. The Atankwidi catchment is a rural and traditional catchment where informal institutions or customary water practices encapsulating culture, social relations, and networks predominantly regulate groundwater irrigation. Local knowledge is the main driver of groundwater irrigation in the catchment. Guided by this, assessing the informal institutions may provide important information on the planning, implementing, managing, and operating of MAR projects in a better way.

The overall aim of this study is thus to provide information about institutional guidelines that need consideration by proponents of MAR who may wish to undertake MAR for irrigational purposes in a developing country such as Ghana. The paper starts by looking at issues surrounding groundwater dry-season irrigation in the Atankwidi catchment of northern Ghana. We further review the literature about MAR globally and in Ghana, and offer an insight into the characteristics of the Atankwidi catchment and the source of data for the study. A presentation on both formal government- and catchment-level traditional/informal institutions constitutes the next part of the discussion. The paper concludes by recommending a comprehensive study on the storage capacity of all the aquifers in the catchment. There is also a need for a detailed quantitative study on the actual groundwater demand for dry-season irrigation in the catchment.

## 2. Review of the Existing Literature

### 2.1. Managed Aquifer Recharge

According to Dillon [23], MAR refers to the purposeful recharge of water to aquifers for subsequent recovery or environmental benefits. According to Gale [17], MAR projects enable the storage of water in aquifers for subsequent use, to increase groundwater levels, improve water quality, address saline intrusion in waters, etc. Asano and Cotruvo [18] revealed that artificial recharge methods (including MAR) are able to reverse a decline in groundwater levels, enhance groundwater quality by preventing saltwater intrusion, and enable the storage of reclaimed municipal wastewater for future use. Dillon [23] argues that MAR can harvest and reuse water, citing the City of Mount Gambier in Australia where drainage wells have replenished a karstic aquifer for 120 years without indications of poor water quality. MAR also offers an opportunity to build groundwater mounds which block the inflow of contaminated

water from areas upstream [24]. Bouwer [25] explained that artificial recharge can address problems of land subsidence, store water, and enhance water quality through soil aquifer treatment or geo-purification, among many other benefits. It is further realized that ponded infiltration (during percolation through the vadose zone and passage through aquifer) can improve the quality of treated sewage effluent or enhance the quality of surface water for irrigation purposes [26]. There is extensive literature on other classifications of MAR [2,3,17,23]. Table 1 provides some highlights of MAR [5,27].

**Table 1.** Managed Aquifer Recharge (MAR) methods.

| Main MAR Methods | Specific MAR Methods | |
|---|---|---|
| Techniques referring primarily to getting water infiltrated | Well, shaft, and borehole recharge | Aquifer Storage and Recovery (ASR)/Aquifer Storage, Transfer and Recovery (ASTR) shallow wells/shaft/pit infiltration |
| | Spreading methods | Infiltration ponds and basin Flooding, Ditch, furrow, drains irrigation |
| | Induced bank infiltration | River/lake bank filtration Dune filtration |
| Techniques referring primarily to intercepting the water | In-channel modifications | Recharge dams, Subsurface dams, Sand dams, Channel spreading |
| | Runoff harvesting | Rooftop rainwater harvesting Barriers and bunds Trenches |

**Source:** Adapted from International Groundwater Research Assessment Centre (IGRAC) [5] and Ringleb et al. [27].

### 2.2. MAR in Northern Ghana

MAR schemes are noted to be emerging in the northern part of the country. The primary goal of these schemes is usually to make water available for agricultural purposes in the dry season. Some of the communities where these schemes are located are flood-prone areas that usually experience water scarcity, particularly in the dry season. The beneficiary communities are also poor in terms of socio-economic development. A major characteristic of these MAR schemes is that they are usually combined with or augmented by other technologies in developing groundwater for use. These schemes have also been implemented by either individuals or agencies outside the domain of formal government agencies. The Kpaloworgu Sand Dam in the Upper West region, the first of its kind, was initiated by non-Ghanaian individuals. The Aquifer Storage and Recharge (ASR) and Pit Infiltration MAR schemes were implemented by institutes/agencies, not directly connected to formal government institutions.

Regarding the ASR method, its schemes have been implemented by the International Water Management Institute (IWMI) as part of the "Securing Water for Improved Seed and High-Value Vegetable Production in Flood-Prone Areas of Northern Ghana" (Secure Water) scheme. The ASR method is supported by the Bhungroo technology (an an Indian indigenous water-harvesting technique) which, according to Owusu et al. [28], allows for the harvesting of excess floodwater for agricultural use during the dry season. The technology encompasses the harvesting, storage, and abstraction of water. The technology comes with the lifting of water from the Bhungroo and putting it into an overhead tank for on-site water storage and distribution, and a drip or sprinkler irrigation system for applying water to crops [28]. The Jagsi and Kpasenkpe communities in the West Mamprusi District in the Northern Region and the Weisi community in Builsa South District in the Upper East region are the beneficiary communities. These are flood-prone communities that experience floods annually.

The Pit Infiltration with PAVE technology is found only in the northern region of Ghana and adopted by Conservation Alliance International (CA). PAVE Irrigation Technology is a German-originated rainwater harvesting and aquifer recharge irrigation system that injects excess water underground during periods of rainy days and floods. The technology captures flood water and filters and injects into the aquifer and unsaturated fractures. The water is stored underground, from which farmers can use it for about six months. Simple pumps are used to tap the water from

the pits. Water is extracted from the injection pipe, or through an alternative pipe. Savelugu Nanton Municipal Assembly, Tolon-Kumbungu, and the West Mamprusi districts constitute the beneficiary districts. There are thirteen projects.

## 2.3. Institutions and their Nature

Definitions of institutions come from different sources, each reflecting the backgrounds or perspectives of the theorists. A commonly agreed definition of institutions is thus lacking [29,30]. As a result, one finds a plethora of them in existence, with new ones still emerging. Common among these definitions is the one offered by North [31] which states that institutions are "the rules of the game in a society". Saleth [32] added that institutions are the rules, norms, and strategies which guide the activities and behaviour of individuals. Within the domain of water resources, institutions are the "rules that define action situations, delineate action sets, provide incentives, and determine outcomes . . . in the context of water development, allocation, use, and management" [32].

In discussing their nature, Helmke and Levitsky [33] explained that institutions are both formal and informal rules and procedures that structure social interactions. To simply this, they defined informal institutions as "socially shared rules, usually unwritten, that are created, communicated, and enforced outside of officially sanctioned channels. By contrast, formal institutions are rules and procedures that are created, communicated, and enforced through channels widely accepted as official" [33]. Rauf [34] is of a similar view that institutions are both formal and informal, with the informal institutions in the form of norms, customs, and traditions having the additional advantage of generating social capital and influencing resource utilization among people. Institutions are both formal and informal structures; they are often multi-purpose, intermittent, and semi-opaque in operation, and not consciously designed [35]. Focusing on formal institutions, Saleth and Dinar [36] decomposed formal institutions with regard to water resources as laws, policies, and administrative structures.

Concerning their place within the domain of water resources, Kemper [37] discussed that institutions define groundwater instruments, which deal with user rights, abstraction permits or concessions, groundwater tariffs, and subsidies, as they even create groundwater markets. Vatn [30] also noted that institutions define access or ownership of given natural resources. Gale [17] suggested that in examining institutions in relation to MAR, attention should be given to water rights, land ownership, legal and regulatory issues, etc.

In Ghana, Fuest et al. [38] noted that institutions are in the form of statutory laws, legal instruments and regulations, national policies, by-laws, local laws, and project laws. In tracing their source, as far as groundwater use for irrigation is concerned, one sees them emanating from two levels: the national and local levels. As a country where decentralization is practised, the Local Government Act 1993 Act 462 mandates District/Municipal/Metropolitan Assemblies to make by-laws and take certain decisions for the purpose of a function conferred on them. Therefore, local entities like the Assemblies, based on their local conditions, make by-laws to complement those at the national level. Nonetheless, groundwater instruments have not been designed and used within the domain of groundwater irrigation in the country. Occasions that call for securing permits for the use of groundwater for irrigation by smallholder farmers are also uncommon.

In view of all these things, this paper chooses to define institutions as both formal and informal structures where the former comprises policies, laws/legislative instruments, and administrative structures, while the latter encompasses norms, taboos, traditions, etc. which operate outside officially sanctioned channels. These institutions serve multiple purposes and are consciously or unconsciously developed.

## 3. Materials and Methods

### 3.1. Study Area

The study area of choice was the Atankwidi catchment, a tributary of the White Volta Basin (Figure 1). It is transboundary in nature and covers an area of about 286 km$^2$. The portion of the

catchment in Ghana is about 159km$^2$ [39]. Its population in 2010 was 45,841 [40]. The catchment in Ghana is located in the Upper East region, of which four Districts/Municipalities—namely, the Kasena/Nankana Municipality, Kasena/Nankana East District, Bolgatanga Municipality, and Bongo District—are its local political units. The catchment covers six communities in Ghana: Kandiga, Sirigu, Yuwa, Zorko, parts of Sumbrugu, and Mirigu (Figure 1). The people in these communities speak the same language, with similar ethnic characteristics. Agriculture is their major economic activity. Sumbrungu is the most populous and more urbanized community among them.

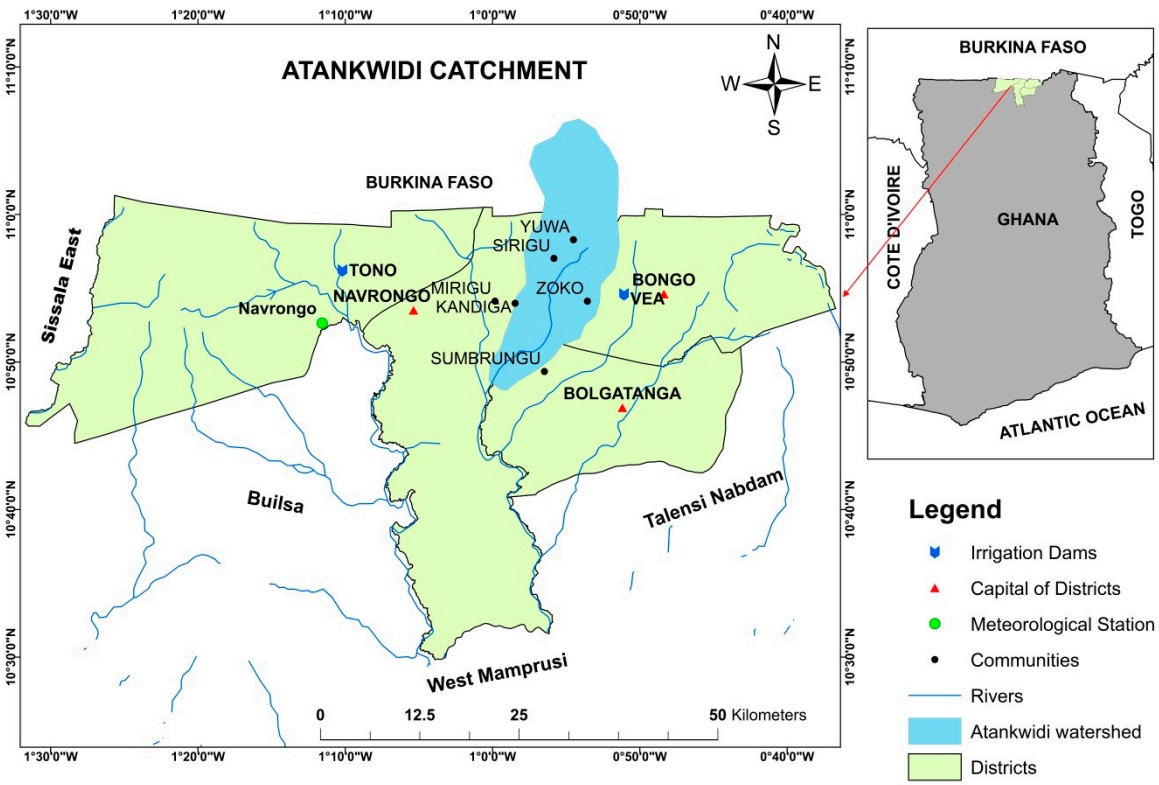

**Figure 1.** Atankwidi catchment.

The catchment was chosen because the people here rely almost entirely on groundwater for their livelihood activities. It remains a major catchment in the northern part of the country, where groundwater irrigation is significant in the dry season.

The climatic conditions of the catchment, according to Barry et al. [8] are that of the Sudan-Savanna zone, associated with high temperatures and a mono-modal rainfall distribution, with a distinct rainy season lasting from approximately May to September. In examining the entire Volta Basin of which the Atankwidi catchment forms part of, Amisigo [41] stated that the basin is characterized by two main geological systems: the Precambrian platform and a sediment layer, and the Voltain system (which covers about 45% of Ghana). The basin is also noted for the absence of primary porosity; therefore, groundwater occurrence in most of the basin is through the development of secondary porosity. The aquifer systems in the basin (including its portion in Ghana) are highly discontinuous with groundwater, occurring mostly under semi-confined or leaky conditions [42]. Specifically, on the Atankwidi catchment, Martin [11] noted that a larger part of the catchment is associated with the presence of the Paleoproterozoic granitoids. Also, faulting activities have resulted in two smaller shearing faults which are found in the northern tip of the area in the west-east direction.

There are three aquifers characterizing the catchment [11]. These are the discontinuous, shallow, perched aquifer, the regolith aquifer, and the fractured aquifer (Figure 2). Among these three, the regolith aquifer constitutes the principal aquifer in the weathered mantle, resulting in a continuous aquifer whose average saturated thickness is 25 m and hydraulic conductivity being from 2.5E-6 to

2.5-5E m/s, which supplies the yield of most boreholes. The discontinuous shallow perched aquifer is characterized by coarse soils of 0.5 m to 1.5 m thickness and covered by a less permeable clayey or lateritic layer. Nonetheless, this aquifer provides water to traditional wells at very shallow depth, even though they dry up in the dry season.

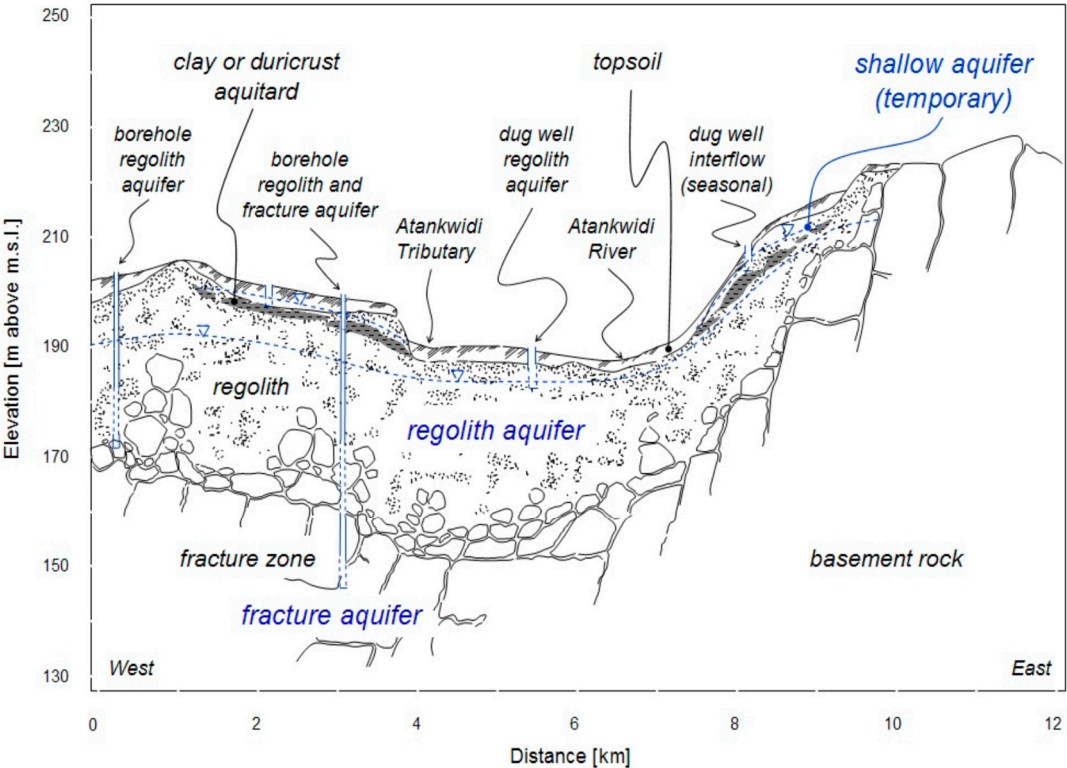

**Figure 2.** Hydrogeological cross-section of the Atankwidi catchment (Source: Martin [11]).

Focusing on the Volta basin (of which the Atankwidi catchment forms part of), Amisigo [41] observed that the mean monthly potential evapotranspiration exceeds the mean monthly rainfall for most of the year in the basin. Regarding recharge, Namara et al. [16] stated that recharge in the Volta Basin is highly variable, both spatially and temporally, and also low when compared with annual rainfall and evapotranspiration. According to Martin [11], groundwater recharge specifically in the Atankwidi catchment is between 1 and 13% of the mean annual rainfall, which is about 990 mm.

Groundwater quality in the entire Volta basin presents health concerns, especially with regard to the high level of fluoride concentration found in the central part of the basin in northern Ghana [16]. Nonetheless, the water quality is considered suitable for irrigation [43]. The total irrigable area by groundwater, as documented by Barry et al. [8] is about 387 ha.

According to Kwoyiga and Stefan [15], groundwater irrigation is largely undertaken by men. These farmers draw from local knowledge to explore and exploit groundwater for irrigation. Farmers rely on themselves, family members, and friends for financial support and to provide security on their farms. The sizes of the farms are different, and a farm may depend on 2–3 wells for water. Wells are either shallow as in the riverine or deep when on the field. Water is extracted using buckets, ropes, and pumping machines. The crops grown are mostly vegetables like onion, spinach, pepper, tomatoes, cabbage, lettuce, etc. Farmers market their crops either on their farms or in the nearby markets.

### 3.2. Data for the Study

In order to have an in-depth knowledge and understanding, and also to support proponents of MAR with information about the institutions that need consideration when implementing MAR for irrigation purposes in a developing country such as Ghana, a qualitative approach was employed. In

each of the six study communities, two focus-group discussions (each group comprised six members) were held with farmers. Four key informants, such as chiefs, elders, and *tindana* (earth priest) in each community were contacted to obtain data about institutions that regulate groundwater development, as well as use in the catchment. Two officials from the Water Resources Commission (WRC) and one official from the Environmental Protection Agency (EPA) in the Upper East region were also contacted to obtain information about the institutions and management support that may aid MAR implementation.

All questions were open-ended. This was to give respondents the opportunity to freely respond to questions without being limited or influenced. Moreover, there was no pre-existing information about the informal institutions in the catchment—hence the need to avoid close-ended questions. Interviews with the people at the catchment were done using interview guides because most of the people there were illiterate. Questionnaires were used to get responses from the officials at the WRC in the Upper East region. An interview guide was used during the interview with the official at the EPA, which was a decision taken by the official.

All respondents in this category were purposefully selected. Thus, a four-month field trip was undertaken in Ghana in 2017 and a two-month field trip in 2018.

Formal discussions with three purposefully selected members of the Innovative web-based Decision Support System for Water Sustainability under a Changing Climate (INOWAS) Junior Research Group in Technische Universität Dresden, TU Dresden, Germany, together with secondary data from this group about MAR methods, applications, and projects yielded supporting data. A review of the policy documents and legal and institutional frameworks for providing groundwater in Ghana was carried out. These documents were accessed from the internet, as well as from the offices of the WRC and EPA in Bolgatanga.

The first author who collected the data relied on an interpreter who doubled up as a research assistant during the interviews. The data collection tools employed were interviews, questionnaires, observation, conversations, and informal discussions. Data were first recorded, transcribed, and then analyzed manually.

## 4. Results and Discussion

Drawing from the empirical data and extensive analysis of existing literature and policy documents from Ghana, the ensuing presentations look at the various institutions in the country that need to be considered prior to and during the implementation of MAR methods for irrigational purposes in the Atankwidi catchment.

### 4.1. Formal Government Institutions

A review of Ghana's official documents and secondary data, together with interview responses of officials of the WRC and the EPA provided the following results. The results here show the various relevant formal government institutions designed at both national and District/Municipal (local) governments. However, the presentation here about the list of relevant institutions is inexhaustible, since new ones are emerging while some existing ones are being modified.

#### 4.1.1. Laws/Legislative Instruments

Currently, there is no legal provision/rule in Ghana that prohibits MAR implementation, and neither is there any single legislation which states that MAR should be implemented to boost groundwater resources. However, some existing legislation or legal instruments appear more explicit and relevant, which proponents of MAR projects need to consider.

- The 1992 constitution of the Republic of Ghana

Land is an important factor as far as MAR is concerned. In Ghana, land and groundwater appear inseparable (an interpretation of the local people). Land ownership in the country has changed over

the years; however, a final decision was taken in the 1990s. The 1992 constitution (Article 36(8)) vests all customary lands in the appropriate stool, skin, or landowning family on behalf of and in trust for their people, to be managed as pertained in the duties of the traditional authorities based on customary law. In the Atankwidi catchment, the land is considered skin land. The land is owned by the community, families, or clans. It is entrusted in the various earth priests called *tindana,* who act as the mediator between the gods and the people. This is important for engineers and proponents for MAR when deciding on the source and location of MAR project. Land for such a purpose must be negotiated with and obtained from the local people.

- The Water Resources Commission Act 1996 (Act 522)

The act stipulates that regardless of geographic regions, all water resources belong to the state to be held in trust by the president of the Republic of Ghana. This connotes that groundwater resources associated with MAR form part of the country's pool of water resources, whose ownership is vested in the President of the Republic. This is important for understanding groundwater rights and their enforcement in the catchment, as far as MAR schemes are concerned.

- The Drilling Licence and Groundwater Development Legislative Instrument (L.I) 1827

This Act mandates engineers to obtain drilling licences from the Water Resources Commission. Before the project commences, there is a need for written notification to the Water Resources Commission for permission. It is thus required of MAR engineers to acquire drilling licences beforehand. Wells drillers, as part of the MAR process, are also cautioned by this act to make sure that wells drilled do not pose threats in the form of pollution or contamination to groundwater aquifers.

- Water Use Regulations (L. I) 1692: 2001

MAR goes beyond some of the exemptions of the Water Use Regulations (L. I) 1692: 2001. The regulations make exemptions to people who: apply manual means to lift groundwater from wells; persons who intend to abstract water through mechanical means where the abstraction level is below 5 L/s; and farmers who use the water to cultivate areas of land that do not exceed 1 h. These water users are, however, obliged to register their activities with the District Assembly. Depending on the scale of the MAR scheme, it is important for users of the water to obtain permits, especially if is for irrigation on a large scale.

- The Rivers Act, 1903

This Act states that "a person shall not, without a licence from the Minister, pump, divert, or by any means cause water to flow from a river (a) for purposes of irrigation, or for mines, factories, or any other commercial or industrial purposes, or (b) to generate power". The source of water for MAR includes rivers and other surface water bodies. It is, therefore, compulsory for MAR engineers or proponents to get permits or licences in a situation where the source of water for MAR (river bank infiltration method) is from a river, like the Atankwidi River.

- The Dam Safety Regulations, 2016

These regulations state that licence is required to construct, alter, operate, conduct, or decommission a dam. Therefore, should any MAR methods (subsurface dams or sand dams), be desired, registration of the project is deemed necessary, after which the Dam Safety Licence is issued by the Dam Safety Commission before the project kick-starts.

- Environmental Assessment Regulations, 1999

Regardless of the scale of MAR, an Environmental Impact Assessment may be required, as stipulated in the Environmental Assessment Regulations, 1999. These regulations stipulate that the Environmental Impact Assessment is mandatory for groundwater development for industrial, agricultural, or urban purposes.

From the results, it is realized that there exist a plethora of laws/legislation relating to MAR in Ghana. However, there are no applicable by-laws of the four Districts/Municipalities in this regard. A critical look at the existing legislation shows that they are designed to serve multiple purposes, which confirms the argument of Cleaver [44] about the nature of institutions in Africa. These institutions, in view of MAR, touched on issues such as the source of water for recharge, land, and water rights, construction/drilling licences, environmental permits, and use of the water. They are legally binding and may hinder MAR implementation if the approval requirements are not met. However, there is missing information concerning the specific guidelines for water treatment (quality) for recharge, so MAR operators may, therefore, need to adopt certain international guidelines. There is also no information on the preferred MAR methods in the country. Groundwater instruments, as far as irrigation is concerned, are also not designed. Even though local farmers, based on local knowledge, conjunctively use surface water and groundwater, this has not been spelt out by these institutions. Demand management is only applicable to groundwater for domestic purposes. These issues are important, and may negatively affect the implementation and operations of MAR if not properly considered. Therefore, it behoves MAR operators to consider these before initiating MAR projects. Like Casanova et al. [45] noted of France, and Megdal et al.'s [46] discussion of the USA state of Arizona, Ghana's laws/legislation, to some extent, are explicit on the requirements that need to be met in order for MAR to be accepted. When compared with policies and administrative agencies/organisations, it can be concluded that the relevant existing laws provide better and more comprehensive information for MAR activities in the country, and failure to comply with them may result in the rejection of MAR projects.

### 4.1.2. Policies

Specific policy provisions for MAR as part of groundwater resource development are lacking. Nonetheless, the following policies provide information relating to MAR implementation.

- Groundwater Development Strategy, 2011

This strategy, though yet to be implemented, may contribute to the realization of MAR objectives. The strategy intends to boost data and information on groundwater, strengthen capacity in terms of technical and organisational aspects, and encourage stakeholder participation, among other things. When implemented, this strategy may reduce the burden of engineers or proponents of MAR schemes who will need to build the capacities of the organization or agencies to manage and operate the schemes.

- The National Rainwater Harvesting Strategy, 2011

Formulated by the then Ministry of Water Resources, Works, and Housing (MWRWH) as a roadmap to augment water service delivery in both rural and urban areas of the country, the strategy spans a period of 2012–2025. It concerns rainwater harvesting, which may be relevant regarding the source of water for MAR projects.

- The Ghana Climate Change Strategy, 2012

This strategy advocates for methods of improving water resources for agricultural activities. It implies that in this era of climate change and its impacts, MAR may be considered a part of the country's adaptation measures.

- The Ghana National Climate Change Policy, 2014

This policy promotes the constructions of water storage systems through rainwater harvesting. Water harvesting in this source could boost the availability of water sources for MAR.

- The Ghana Water Policy, 2007

This policy also encourages Ghanaians to harvest rainwater at the household level and that of the community level. This is to boost water availability in general by broadening water sources, and is

also a way of containing excess water on the ground surface. This is a plus towards the realization of MAR, as rainwater harvesting could serve as the source of water for recharge.

- The National Environmental Policy, 2012

This policy lends credence to the need to subject water resources development projects to an Environmental Impact Assessment. As per this policy, MAR projects are therefore supposed to undergo impact assessment to attain the nature of impacts that MAR may have on the environment.

In view of policies, issues such as the source of water for MAR and organizational management of the water, as well as awareness/education about MAR activities have been captured. However, these are only general and are not action-oriented, as far as MAR is concerned. None of these policies specifically made provisions for groundwater development through recharge. A critical look at the climate change adaptation policies reveals that the aim of harvesting surface water in this regard is to avert hydrological disasters (flood control) and not to capture water to deliberately boost recharge. The National Rainwater Harvesting Strategy and the Ghana Water Policy, 2007 have been put in place as part of measures to boost household and municipal water supply, but not for groundwater recharge. One would have expected that the other related policies in the country, such as the Food and Agricultural Sector Development Policy (FASDEP I& II) of 2003 and 2007, respectively; the Ghana Water Policy, 2007; the National Climate Change Adaptation Strategy, 2012; and National Climate Change Adaptation Policy, 2014 would have prioritized and included groundwater resources development (including recharge) in them—unfortunately, none of these did.

In that same direction, the Ghana Irrigation Development Policy 2011, in section 5.3.1, skeletally stated that efforts shall be made to promote access to safer groundwater or safer irrigation practices where only marginal-quality water is available. The state of the Water Policy 2007 regarding groundwater irrigation is but vague. This is because the policy in section 2.2.3 (Water for Food Security) only touches on supporting micro-irrigation schemes among rural areas without specifying the source of water for these schemes. Generally, water resource policies outline the roles of the government and other stakeholders, define monitoring and controlling measures, and state how to build capacity for management. Unfortunately, a review of the existing policies shows that groundwater development through MAR is completely absent within the policy framework of the country. This may negatively affect the adoption of MAR in terms of resources in Ghana. Proponents of MAR will need to identify the ways and means of addressing these constraints when planning MAR schemes.

### 4.1.3. Administration

There are some administrative bodies in the country that deal with water resources in general on one hand, and with irrigation on the other. The activities of the following agencies nonetheless relate to MAR and groundwater.

- Water Resources Commission (WRC)

This agency has representative offices at the regional level. The commission processes all water rights and permits. It is the lead regulator of all water resources in Ghana. It is therefore important for MAR proponents to involve this commission, especially at the planning stage.

- Environmental Protection Agency (EPA)

This agency is the leading environmental agency of the country's government, with offices in all the regions. It is responsible for conducting Environmental Impact Assessments (EIA). Prior to MAR implementation, notification to this agency is required.

- Water Research Institute (WRI)

This institute, unfortunately, has no offices at the regional level, as it is only based in Accra, the capital. It is relevant for MAR projects because this outfit has conducted research and collaborated

with other research organisations concerning information-gathering pertaining to the hydro-geology, hydrochemistry, and other issues about groundwater for the entire country.

- District Assemblies (DA)

The assemblies are decentralized agencies or local government entities who play both political and administrative roles at the local level. The assemblies, as part of their responsibilities, are supposed to monitor drilling and wells construction activities in the district. MAR wells constructors will, therefore, need to register with the assembly.

- River Basin Management Boards

These boards have been purposefully constituted to manage river basins in the country. Unfortunately, their focus is more on surface water than groundwater resources. Despite this, they are regarded as important stakeholders whose platform enables the management of water resources, as well as for conflict resolution.

The critical issues raised here include responsibilities about groundwater regulation, pollution, and hydrogeological knowledge/information. The WRC only regulates, rather than develops [47], and from the interviews, it is realized that its responsibilities, especially at the regional level, focus more on surface water than on groundwater resources. At the moment, MAR implementation may be constrained due to the absence of technical, financial, and management support from these institutions. For instance, it is difficult to identify people with technical skills and resources available to design, construct, and operate MAR projects.

Groundwater development and management, which should have been a prerogative of the Irrigation Development Authority according to the Irrigation Development Authority Act, 1977, has not been carried out (see Ministry of Food and Agriculture [48]). The District/Municipal Assemblies only support the management of groundwater facilities for domestic purposes. This implies that currently, and as far as groundwater development for irrigation is concerned, MAR projects cannot be associated with any administrative agency in the country. It is not surprising that the MAR projects in the northern part of the country are spearheaded and championed largely by agencies such as the Conservation Alliance and the International Water Management Institute, together with the local communities. As such, MAR proponents and engineers should be willing to develop groundwater with limited support from these agencies. Proponents should also identify individuals or agencies that may be responsible for managing the MAR projects when implemented. This can be done through collaboration with the WRC, other water agencies in the country, or through the creation of an independent organization for such a purpose.

### 4.2. Catchment-Level Institutions

From the interviews conducted in the six communities in Atankwidi, it has been revealed that informal institutions remain strong and influential as far as groundwater irrigation is concerned in the catchment.

Nature of Institutions

The Atankwidi catchment, as noted already, is a rural catchment where informal institutions are in full operation. These are limited largely to the catchment and the neighbouring communities who are connected through kinship or the social network. These informal institutions are in the form of:

- Taboos
- Rules
- Customs/norms/traditions
- Groundwater leaders

In order to highlight the relevance and provisions of these institutions in respect of MAR, a detailed presentation is shown in Table 2. This takes into consideration the regulatory aspect, approval requirements, and agencies/organizations in regard to MAR.

**Table 2.** Catchment-level institutions, and their provisions for MAR.

| Institution | Provision for MAR |
|---|---|
| Rules<br>1. Land and groundwater are inseparable gifts from the gods/nature and therefore belong to every member of the catchment; thus, access is free.<br>2. Extraction of groundwater is mostly the responsibility of individual farmers.<br>3. Farmers exploit groundwater through the construction and maintenance of individual wells. No one is supposed to trespass on another irrigator's land/groundwater. Irrigators, however, sometimes help one another by granting free access to water in their wells.<br>4. Wells can be constructed at any time of the year.<br>5. Information about the spiritual component of irrigation is kept secret among irrigators. | 1. Land/groundwater here is owned by the community; thus, land for siting the MAR scheme must be acquired from the local people.<br>2. MAR projects can be individually or communally owned. This is important for managing and operating MAR projects.<br>3. Farmers may be able to operate MAR projects (small-scale) individually; however, social network/capital may influence the process, depending on the nature of the MAR project.<br>4. MAR projects can be implemented at any time of the year.<br>5. Local knowledge of farmers about groundwater offers a quicker way to understanding and managing the MAR projects. |
| Taboos<br>1. There shall be no construction of wells in sacred groves or places considered sacred.<br>2. There shall be no fetching of water at night, especially near places considered to be the abodes of the gods. | 1. The physical location of the MAR scheme must be accepted and approved by the local people.<br>2. As a rural catchment where there are no wastewater treatment plants, the source of water will obviously not come from a wastewater treatment plant. Therefore, where the source of water for the MAR scheme is from a river, detailed community consultations (chiefs/earth priests) are required prior to implementation. |
| Customs/Norms/Traditions<br>1. All water resources are sacred.<br>2. Rituals and sacrifices are made for abundant water (rains to recharge). | 1. Water from MAR projects is equally considered sacred and treated diligently.<br>2. This is important for conserving and protecting MAR projects, due to the importance attached to all water resources in the catchment. |
| Groundwater leaders<br>Chiefs, elders, heads of clans/families, farmer groups/associations, and youth groups. | These are local but traditional political leaders of the catchment, gate keepers, and major decision-makers. They will contribute to managing and operating MAR projects. Their participation in decisions regarding the scheme must always be considered. |

The catchment-level institutions, as far as groundwater irrigation is concerned, are informal. They are undocumented and enshrined largely in the belief systems and traditions of the people, as described by Helmke and Levitsky [33]. They are in the form of taboos, traditions, and norms, and many others are embedded in the local knowledge of the people, which is applied in groundwater irrigation in the catchment. They are also local in nature in that they are only limited to the catchment. These institutions have not been consciously developed for groundwater irrigation purposes only. They have historical connotations, are enmeshed in the people's culture, and are thus not easily altered. Social relations and networks penetrate these institutions. The satisfaction derived from the application of these institutions is not necessarily expressed in economic gains, but is more about social welfare. Their relevance has been noted by Obeng-Odoom [49] in the entire water sector of Ghana in the past.

Fortunately, they have some provisions for MAR (Table 2), especially in relation to planning and operating MAR projects. These institutions provide information on authorization, the approval of sites, and the location of MAR projects. They also define water rights. These institutions further spell out the leadership structure in the catchment that may be crucial for the governance, management, and operation of MAR schemes. Every farmer already possesses knowledge of groundwater, which is significant for building/improving human resource capacities to operate MAR projects.

It can be said that these institutions do not prohibit, but rather favour the adoption of MAR in the catchment. However, choices/decisions of the local people must be respected. For instance, any scheme that fails to take into consideration the taboos, customs, and traditions of the local people will be resisted and may not be implemented.

## 5. Conclusions

The increasing growth rate of the population, fallen groundwater tables, impacts of climate change on water resources, and the booming of groundwater irrigation may constitute the basis for more MAR activities in Ghana. Moreover, although developing countries like Ghana have no specific institutions regarding artificial methods of groundwater recharge, the existing ones do not prohibit it.

Formal government institutions, like laws or legislative instruments, provide sufficient but relevant information on the requirements and regulatory structures for MAR schemes in Ghana, even though policy formulations failed to significantly capture groundwater development through artificial methods. Policy support at the moment may negatively impact the adoption and operation of MAR projects in the country. Proponents of MAR will, therefore, need to mobilize a lot of resources (technical, financial, and managerial) in order to achieve MAR goals in the country. Catchment-level institutions, in the form of rules, taboos, customs, and practices, favor MAR adoption through local knowledge, planning, and management and operations. All institutions must thus be given significant attention to facilitate the approval, easy acceptance, implementation, and operation of MAR projects.

For MAR to be effectively adopted in Atankwidi, this paper recommends a comprehensive study of the storage capacity to be done of all the aquifers in the catchment. Currently, there is no available study about the storage capacity of all the aquifers in the catchment; the only study at the moment is by Barry et al. [8] who studied only one of the aquifers. There is also the need for a detailed quantitative study about the actual groundwater demand for dry-season irrigation in the catchment, since the information at the moment is largely qualitative. Land use is another issue that needs consideration. Environmental degradation, population growth, and agricultural activities continuously alter the environment. Land-use studies have been done already, but this information needs to be updated so as to map out areas that may be suitable for MAR.

**Author Contributions:** Conceptualization; Data curation; Formal analysis; Funding acquisition; Methodology and Writing–original draft were by L.K.; while Resources, Supervision; Writing–review & editing, were done by C.S.

**Funding:** The research received funding from the Government of Ghana through DAAD (Germany) and the Graduate Academy, TU Dresden for the doctoral studies/field work in Ghana. The University for Development Studies (UDS), Ghana also supported this study financially.

**Acknowledgments:** The authors acknowledge the contributions of Paul Alagidede through informal discussions to the article. The INOWAS Junior Research Group, TU Dresden is also acknowledged for its support in terms of literature/materials for the article. We further acknowledge the assistance of Mr Francis Anafo during the data collection in Ghana. We thank the people of the Atankwidi catchment and the officials of the various organisations/agencies in Ghana for their responses and materials. The authors again, wish to thank the anonymous reviewers for carefully reviewing the article.

**Conflicts of Interest:** The authors declare that there is no conflict of interest.

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
