# Peer review of "Institutional Feasibility of Managed Aquifer Recharge in Northeast Ghana"

_sustainability, doi:10.3390/su11020379_

Round 1
Reviewer 1 Report
The manuscript presents an interesting overview about legal and policy background of artificial groundwater recharge in Ghana in general and in a particular catchment. Besides the summary of formal and informal institutions, laws rules and norms, interviews are considered to assess the current situation.
The following questions remain unclear to me and I strongly recommend to add some detailed information and interpretation regarding these aspects:
what is the difference between national and local scale in this context?
for what reason did you choose this particular catchment - what is special here, what is not comparable or on the other side representative for the total country
Did you use standard questions for all interview partners, did you use a questionaire, did you use standard methods for evaluation? Present the design of your filed work more detailed!
Due to the fact that these informations are missing in the current version, the scientific approach seems rather weak. However, I´m confident that it could be improved with only answering these questions very easily. Thus, I classify the suggested improvements as minor revisions.
Author Response
The manuscript presents an interesting overview about legal and policy background of artificial groundwater recharge in Ghana in general and in a particular catchment. Besides the summary of formal and informal institutions, laws rules and norms, interviews are considered to assess the current situation.
The following questions remain unclear to me and I strongly recommend to add some detailed information and interpretation regarding these aspects:
what is the difference between national and local scale in this context?
Response
National refers to country wide or nation state and local means the District/Municipality (Ghana practises some form of decentralisation).
for what reason did you choose this particular catchment - what is special here, what is not comparable or on the other side representative for the total country.
Response
The suggestion has been noted and considered in the section on study area
Did you use standard questions for all interview partners, did you use a questionaire, did you use standard methods for evaluation? Present the design of your filed work more detailed!
Response
We did not use standard questions because different data was required from different respondents. Questionnaires and interview guides were used. All questions were open ended.
This suggestion has been considered also in the revised version
Reviewer 2 Report
GENERAL COMMENTS
This manuscript focused on “Exploring Institutional Provisions for Managed Aquifer Recharge to boost Groundwater for Dry Season Irrigation in North East Ghana”. This topic is related to the scope of this journal. The authors attempt to explore the feasibility of implementing Managed Aquifer Recharge (MAR) in the catchment as a way of boosting groundwater based on an institutional (Ghana) insight. However, this manuscript should be systematically revised before it can be accepted. The following comments may help to improve the present work.
v The objectives of the present study are merely to elucidate (i) what provisions do formal government institutions have for MAR? (ii) What catchment level institutions may relate to MAR in Ghana. In other words, the significance of this study to engineering practice is not obvious.
v Moreover, a plethora of institutional dispositions (Legislative Instruments, Policies, and Administration) regarding the application of MAR is presented. Here again, what is the message? Additionally, information is provided without any critical analysis.
v What impact the formal government institutions do (or do not) have on the development of MAR? To what extend the current institutions support (or not) to the boosting of MAR.
v Recharge is a useful measure for control of land subsidence and safety, e.g. (Shen and Xu 2011; Wu et al., 2016, 2017). Suggest to do a comprehensive review on the benefit of recharge technology.
v Wu, Y.X., et al (2016). Characteristics of dewatering induced drawdown curve under blocking effect of retaining wall in aquifer, Journal of Hydrology, 539(2016), 554-566. doi: 10.1016/j.jhydrol.2016.05.065
v Shen, S.L., and Xu, Y.S. (2011). Numerical evaluation of land subsidence induced by groundwater pumping in Shanghai, Canadian Geotechnical Journal, 48(9), 1378-1392. doi: 10.1139/T11-049
v Wu, Y.X., et al (2017). Semi-analytical solution to pumping test data with barrier, wellbore storage, and partial penetration effects, Engineering Geology, 226, 44-51. doi: 10.1016/j.enggeo.2017.05.011.
v We also suggest performing a critical analysis and showing how the results of the proposed investigation are transposable to analogous or bigger scale conditions/situations.
v The institutional framework for implementation of Managed Aquifer Recharge (MAR) is presented. But as previously mentioned, the objectives of such an approach remain unclear. Another important aspect and not the least is the fact that neither recommendations nor perspectives are discussed.
v Please provide some background information regarding the Atankwidi cathement. In fact the latter is presented in Abstract (Line 12) and introduction (line 52) without being previously defined; which alter the logical flow of the paper.
v The reader believes that the sections 1.1 to 1.3 should constitute another independent section, instead of being included in the introduction section.
v Generally speaking, in the present form, this manuscript looks more like a project report than a research paper.
v Please adopt a precise and concise writing style for a better comprehension of the paper.
OTHERS COMMENTS:
1. Scientific writing
v The title of the present manuscript is too long and somehow fuzzy. Please provide a concise and inclusive title.
v The structure of the introduction should be addressed. Indeed it must: i) Describe the objectives of the paper and their relevance to engineering practice; ii) Highlight how the new information can improve practice; and iii) Outlines the sequence of the paper. Addressing these issues will also contribute to make the significance of this paper more obvious. Also, the first paragraph should be moved to the end of the introduction.
v The authors should also clearly describe the objectives of the paper and their relevance to the engineering practice; highlight how the provided information can improve practice (message); and present the sequence of the paper.
v The authors should be more meticulous and rigorous. The aesthetical aspect, the readability and the quality of the figure should be enhanced; especially figure2.
v Line 188: this sentence is not clear, please revise it.
v Lines 12-13: this sentence is not clear, please revise it.
Author Response
v The objectives of the present study are merely to elucidate (i) what provisions do formal government institutions have for MAR? (ii) What catchment level institutions may relate to MAR in Ghana. In other words, the significance of this study to engineering practice is not obvious.
v Moreover, a plethora of institutional dispositions (Legislative Instruments, Policies, and Administration) regarding the application of MAR is presented. Here again, what is the message? Additionally, information is provided without any critical analysis.
Response
The suggestion has been noted and considered in the revised version
v What impact the formal government institutions do (or do not) have on the development of MAR? To what extend the current institutions support (or not) to the boosting of MAR.
Response
The suggestion has been considered
v Recharge is a useful measure for control of land subsidence and safety, e.g. (Shen and Xu 2011; Wu et al., 2016, 2017). Suggest to do a comprehensive review on the benefit of recharge technology.
Response
The suggestion has been considered
v Wu, Y.X., et al (2016). Characteristics of dewatering induced drawdown curve under blocking effect of retaining wall in aquifer, Journal of Hydrology, 539(2016), 554-566. doi: 10.1016/j.jhydrol.2016.05.065
v Shen, S.L., and Xu, Y.S. (2011). Numerical evaluation of land subsidence induced by groundwater pumping in Shanghai, Canadian Geotechnical Journal, 48(9), 1378-1392. doi: 10.1139/T11-049
v Wu, Y.X., et al (2017). Semi-analytical solution to pumping test data with barrier, wellbore storage, and partial penetration effects, Engineering Geology, 226, 44-51. doi: 10.1016/j.enggeo.2017.05.011.
v We also suggest performing a critical analysis and showing how the results of the proposed investigation are transposable to analogous or bigger scale conditions/situations.
Response
The suggestion has been considered
v The institutional framework for implementation of Managed Aquifer Recharge (MAR) is presented. But as previously mentioned, the objectives of such an approach remain unclear.
Response
The objectives have been revised and elaborated
Another important aspect and not the least is the fact that neither recommendations nor perspectives are discussed.
The suggestion has been considered in the conclusion
v Please provide some background information regarding the Atankwidi cathement. In fact the latter is presented in Abstract (Line 12) and introduction (line 52) without being previously defined; which alter the logical flow of the paper.
Response
The suggestion has been considered in the revised version
v The reader believes that the sections 1.1 to 1.3 should constitute another independent section, instead of being included in the introduction section.
Response
This has been modified
v Generally speaking, in the present form, this manuscript looks more like a project report than a research paper.
v Please adopt a precise and concise writing style for a better comprehension of the paper.
Response
The suggestion has been noted
OTHERS COMMENTS:
1. Scientific writing
v The title of the present manuscript is too long and somehow fuzzy. Please provide a concise and inclusive title.
Response
The title has been considered in the revised version
INSTITUTIONS AND MANAGED AQUIFER RECHARGE IN NORTH EAST GHANA
v The structure of the introduction should be addressed. Indeed it must: i) Describe the objectives of the paper and their relevance to engineering practice; ii) Highlight how the new information can improve practice; and iii) Outlines the sequence of the paper. Addressing these issues will also contribute to make the significance of this paper more obvious. Also, the first paragraph should be moved to the end of the introduction.
Response
This has been considered.
v The authors should also clearly describe the objectives of the paper and their relevance to the engineering practice; highlight how the provided information can improve practice (message); and present the sequence of the paper.
Response
The suggestion has been considered.
v The authors should be more meticulous and rigorous. The aesthetical aspect, the readability and the quality of the figure should be enhanced; especially figure2.
Response
Figure 2 has been revised
v Line 188: this sentence is not clear, please revise it.
Response
The sentence has been revised
v Lines 12-13: this sentence is not clear, please revise it.
Response
The sentence has been revised
Reviewer 3 Report
Kwoyiga and Stefan investigate the feasibility of Managed Aquifer Recharge (MAR) in a catchment in northern Ghana. They used (policy) documents and interviews to obtain the results. They noticed that limited studies focus on the institutional aspect of MAR and demonstrate that Ghana have no specific institutions regrading artificial methods of groundwater recharge but at the same time the existing ones do not prohibit it. They concluded informal institutions at the catchment level need critical consideration for easy acceptance and implementation. They also highlight that any MAR scheme that fails to take into consideration the taboos, customs and traditions of the local people will be not successful.
Although I also believe that institutional aspect of MAR are often more important for successful implementation than aquifer characterizations I have several critical points which should be considered. The methodology is not clear to me. What type of questions were asked. How was the answer reported (multiple-choice), single answer, own or given formulation etc. ? They authors talk about six study communities (Line 246)? Where are these communities? Any differences among them? What was the results of the interviews and analyzed data/documents. Is the results Figure 3? If yes please explain and provide more details. Figure 3 comes a bit out of the blue.
Moreover, how does the authors define “their own definition about institutions and their nature”. I mean many examples are provided but it remains unclear how the authors deal with the definition (and which type they choose).
Although I believe that institutional aspect of MAR are important they authors should briefly provide some information about aquifer characterization and what is need to apply successfully MAR. Otherwise, the institutional aspect might support MAR project, but these projects will fail due to “bad” implementation in the field. Some papers about the methods, benefit of MAR for water quality and quantity are provided below.
They study site description should be more elaborated. They authors mentioned that recharge is 1 to 13% of mean rainfall but the value for mean rainfall is not provided. Also a statement like “ wells ae either shallow …or deep when on field (Line 236)”, does not really help to understand the water balance and associated water issues/problems for the study site. Figure 1 is also the same like in a previous publication of the authors (https://www.mdpi.com/2073-4441/10/12/1724) and should be at last adjusted in this paper.
Table 1 would benefit a lot if the techniques and examples would get another column with some references from the literature. The interested reader might have another look there.
Minor comments:
Line 9: MAR is not only use for irrigation issues
Line 27-32: This section should be transferred to the end of the introduction.
Line 41: space between impacts and []
Section 1.1 please provide more details about the benefits of MAR for water quality and quantity and also provide some ideas about method to investigate. Some useful paper could be:
T. Asano, J.A. Cotruvo Groundwater recharge with reclaimed municipal wastewater: health and regulatory considerations Water Res., 38 (8) (2004), pp. 1941-1951
H. BouwerArtificial recharge of groundwater: hydrogeology and engineering Hydrogeol. J., 10 (1) (2002), pp. 121-142
P. DillonFuture management of aquifer recharge Hydrogeol. J., 13 (1) (2005), pp. 313-316
J. Greskowiak, et al.The impact of variably saturated conditions on hydrogeochemical changes during artificial recharge of groundwater Appl. Geochem., 20 (7) (2005), pp. 1409-1426
Moeck, Christian, Dirk Radny, Andrea Popp, Matthias Brennwald, Sebastian Stoll, Adrian Auckenthaler, Michael Berg, and Mario Schirmer. "Characterization of a managed aquifer recharge system using multiple tracers."Science of the Total Environment 609 (2017): 701-714.
Author Response
Although I also believe that institutional aspect of MAR are often more important for successful implementation than aquifer characterizations I have several critical points which should be considered. The methodology is not clear to me. What type of questions were asked. How was the answer reported (multiple-choice), single answer, own or given formulation etc. ? They authors talk about six study communities (Line 246)? Where are these communities? Any differences among them? What was the results of the interviews and analyzed data/documents. Is the results Figure 3? If yes please explain and provide more details. Figure 3 comes a bit out of the blue.
Response
The suggestions have been noted in the revised version
All questions were Open-ended; there were no multiple choice answers
The six communities are; Mirigu, Sirigu, Kandiga, Sumbrungu, Zorkpo and Yua. These communities are all located in the Upper East Region of Ghana. These communities are spread across 4 local political entities in the form of Districts/Municipalities. This information is considered in the revision
All interview responses have been presented under the section RESULTS. This was followed immediately with some discussions.
Figure 3 has been deleted to avoid confusion.
Moreover, how does the authors define “their own definition about institutions and their nature”. I mean many examples are provided but it remains unclear how the authors deal with the definition (and which type they choose).
Response
It has been defined in the last sentence under the heading INSTITUTIONS AND THEIR NATURE
The paper chooses to define institutions as both formal and informal structures where the former comprises policies, laws/legislative instruments and administrative structures while the latter encompasses norms, taboos, traditions among others, which operate outside officially sanctioned channels. These institutions serve multiple purposes and are consciously or unconsciously developed.
Although I believe that institutional aspect of MAR are important they authors should briefly provide some information about aquifer characterization and what is need to apply successfully MAR. Otherwise, the institutional aspect might support MAR project, but these projects will fail due to “bad” implementation in the field. Some papers about the methods, benefit of MAR for water quality and quantity are provided below.
Response
The suggestion has been considered.
They study site description should be more elaborated. They authors mentioned that recharge is 1 to 13% of mean rainfall but the value for mean rainfall is not provided. Also a statement like “ wells ae either shallow …or deep when on field (Line 236)”, does not really help to understand the water balance and associated water issues/problems for the study site. Figure 1 is also the same like in a previous publication of the authors (https://www.mdpi.com/2073-4441/10/12/1724) and should be at last adjusted in this paper.
Response
The suggestion has been considered. Figure 1 has been revised.
Table 1 would benefit a lot if the techniques and examples would get another column with some references from the literature. The interested reader might have another look there.
Response
It is difficult to alter the table provided by IGRAC. Not only will the table look fragmented but difficult to give it a title. However, the authors before presenting the table provided some sources of literature relating to MAR in the text.
Minor comments:
Line 9: MAR is not only use for irrigation issues
Response
Suggestion is noted
Line 27-32: This section should be transferred to the end of the introduction.
Response
The suggestion is considered
Line 41: space between impacts and []
Section 1.1 please provide more details about the benefits of MAR for water quality and quantity and also provide some ideas about method to investigate. Some useful paper could be:
Response
All suggestions have been considered in the revised version.
Round 2
Reviewer 2 Report
In the response, the authors show only simple "considered". However, you did not show where (page, line) in the manuscript was revised. please give a revision details.
Author Response
In the response, the authors show only simple "considered". However, you did not show where (page, line) in the manuscript was revised. please give a revision details.
This manuscript focused on “Exploring Institutional Provisions for Managed Aquifer Recharge to boost Groundwater for Dry Season Irrigation in North East Ghana”. This topic is related to the scope of this journal. The authors attempt to explore the feasibility of implementing Managed Aquifer Recharge (MAR) in the catchment as a way of boosting groundwater based on an institutional (Ghana) insight. However, this manuscript should be systematically revised before it can be accepted. The following comments may help to improve the present work.
v The objectives of the present study are merely to elucidate (i) what provisions do formal government institutions have for MAR? (ii) What catchment level institutions may relate to MAR in Ghana. In other words, the significance of this study to engineering practice is not obvious.
v Moreover, a plethora of institutional dispositions (Legislative Instruments, Policies, and Administration) regarding the application of MAR is presented. Here again, what is the message? Additionally, information is provided without any critical analysis.
Response
1. The significance of the study has been discussed from line 96-123
v What impact the formal government institutions do (or do not) have on the development of MAR? To what extend the current institutions support (or not) to the boosting of MAR.
Response
Formal institutions have the power to stop MAR projects from being implemented when the guidelines are not taken into consideration. The existing policies have not given attention to groundwater recharge through artificial means. This may negatively affect adoption and implementation in terms of technical and financial support. The existing administrative agencies only regulate but do not develop groundwater through recharge. This means that there is lack of agencies with human resources capable of implementing and managing MAR schemes. It, therefore, behoves MAR engineers to identify or train people to handle such responsibilities. This is seen in the Results and discussion section.
v Recharge is a useful measure for control of land subsidence and safety, e.g. (Shen and Xu 2011; Wu et al., 2016, 2017). Suggest to do a comprehensive review on the benefit of recharge technology.
Response
The scope of the paper at the moment is limited only to institutions. A comprehensive review of recharge technology will be considered in another paper. However, Owusu et al. (2018) have studied recharge technology in some communities in northern Ghana.
v Wu, Y.X., et al (2016). Characteristics of dewatering induced drawdown curve under blocking effect of retaining wall in aquifer, Journal of Hydrology, 539(2016), 554-566. doi: 10.1016/j.jhydrol.2016.05.065
v Shen, S.L., and Xu, Y.S. (2011). Numerical evaluation of land subsidence induced by groundwater pumping in Shanghai, Canadian Geotechnical Journal, 48(9), 1378-1392. doi: 10.1139/T11-049
v Wu, Y.X., et al (2017). Semi-analytical solution to pumping test data with barrier, wellbore storage, and partial penetration effects, Engineering Geology, 226, 44-51. doi: 10.1016/j.enggeo.2017.05.011.
v We also suggest performing a critical analysis and showing how the results of the proposed investigation are transposable to analogous or bigger scale conditions/situations.
Response
The revised version of the paper analytically discussed the situation in Atankwidi as seen in the RESULTS AND DISCUSSION segment. This is found from page 8-13
v The institutional framework for implementation of Managed Aquifer Recharge (MAR) is presented. But as previously mentioned, the objectives of such an approach remain unclear.
Response
The objectives have been revised and elaborated as seen from lines 94-122
Another important aspect and not the least is the fact that neither recommendations nor perspectives are discussed.
Responses
The paper made a recommendation in the conclusion of the paper.
v Please provide some background information regarding the Atankwidi catchment. In fact, the latter is presented in Abstract (Line 12) and introduction (line 52) without being previously defined; which alter the logical flow of the paper.
Response
Detailed information about the catchment has been noted in the section ‘study area’, that is page 6
v The reader believes that the sections 1.1 to 1.3 should constitute another independent section, instead of being included in the introduction section.
Response
This has been modified. The new headings are 2.1 to 2.3
v Generally speaking, in the present form, this manuscript looks more like a project report than a research paper.
v Please adopt a precise and concise writing style for a better comprehension of the paper.
Response
The structure of the paper, its objectives and the presentation of the results have been revised. Figure 1 has been redrawn as seen page 6.
OTHERS COMMENTS:
1. Scientific writing
v The title of the present manuscript is too long and somehow fuzzy. Please provide a concise and inclusive title.
Response
The title has been considered in the revised version. The new title now reads: Institutional feasibility of Manged Aquifer Recharge in North East Ghana.
v The structure of the introduction should be addressed. Indeed it must: i) Describe the objectives of the paper and their relevance to engineering practice; ii) Highlight how the new information can improve practice; and iii) Outlines the sequence of the paper. Addressing these issues will also contribute to make the significance of this paper more obvious. Also, the first paragraph should be moved to the end of the introduction.
v The authors should also clearly describe the objectives of the paper and their relevance to the engineering practice; highlight how the provided information can improve practice (message); and present the sequence of the paper.
Response
These suggestions have been considered in some of the responses above.
v The authors should be more meticulous and rigorous. The aesthetical aspect, the readability and the quality of the figure should be enhanced; especially figure2.
Figure 2 has been re-drawn. The overall structure of the paper has been modified. To avoid confusion, the results and discussion were put together
Response
Figure 2 has been revised as responded already.
v Line 188: this sentence is not clear, please revise it.
Response
In order to avoid confusion and ambiguity, the sentence has been deleted
v Lines 12-13: this sentence is not clear, please revise it.
Response
The sentence has been revised. It attempts to explain that farmers dig wells in riverbeds to extract groundwater. These wells, however, dry up easily rendering farmers without sufficient water especially at the peak of the dry season.
Reviewer 3 Report
The objectives of the paper are better defined and the introduction and benefits of MAR for water quality and quantity is improved. Also the authors improved the study site description. In section 4.1.1 and in the following sections, the gaps regarding MAR in the laws, legislative instruments and policies are nicely discussed. There are still many layout and text issues (I just mention a few below).
Title: maybe the following sentence works better. “institutional feasibility of managed aquifer recharge in north east Ghana.”
Line 11: insufficient groundwater quantity or quality or both?
Line 64: reference before the “.
Line 140: space between reference and [].
Line 260: space between reference and [].
Line 269: 2.5E?? a value is missing
Table 2: different font size
Author Response
The objectives of the paper are better defined and the introduction and benefits of MAR for water quality and quantity is improved. Also the authors improved the study site description. In section 4.1.1 and in the following sections, the gaps regarding MAR in the laws, legislative instruments and policies are nicely discussed. There are still many layout and text issues (I just mention a few below).
Title: maybe the following sentence works better. “institutional feasibility of managed aquifer recharge in north east Ghana.”
Response
The suggestion has been effected as reflected in the current title on page 1
Line 11: insufficient groundwater quantity or quality or both?
Response
Insufficient groundwater quantity is the challenge. The quality of the water for irrigation is considered suitable.
Line 64: reference before the “.
Line 140: space between reference and [].
Line 260: space between reference and [].
Response
All the suggestions have been considered in the text
Line 269: 2.5E?? a value is missing
Response
2.5E-6 to 2.5E-5 m/s.
Table 2: different font size
Response
All the font sizes are now the same.
Round 3
Reviewer 2 Report
no comments. accept.